

# The Climate Generator: Stochastic climate representation for glacial cycle integration

Mohammad Hizbul Bahar Arif[1], Lev Tarasov[2], and Tristan Hauser[3]

[1]Faculty of Engineering and Applied Science, Memorial University, Canada
[2]Department of Physics and Physical Oceanography, Memorial University, Canada
[3]Department of Environmental and Geographical Science, University of Cape Town, South Africa

*Correspondence to:* Lev Tarasov (lev@mun.ca)

**Abstract.** This paper presents a computationally efficient stochastic approach to simulate atmospheric fields (specifically monthly mean temperature and precipitation) on large spatial-temporal scales. In analogy with Weather Generators (WG), the modelling approach can be considered a "Climate Generator" (CG). The CG can also be understood as a field-specific General Circulation climate Model (GCM) emulator. It invokes aspects of spatio-temporal downscaling, in this case mapping

the output of an Energy Balance climate Model (EBM) to that of a higher resolution GCM. The CG produces a synthetic climatology conditioned on various inputs. These inputs include sea level temperature from a fast low-resolution EBM, surface elevation, ice mask, atmospheric concentrations of carbon dioxide, methane, orbital forcing, latitude and longitude. Bayesian Artificial Neural Networks (BANN) are used for nonlinear regression against GCM output over North America, Antarctica and Eurasia.

Herein we detail and validate the methodology. To impose natural variability in the CG (to make the CG indistinguishable from a GCM) stochastic noise is added to each prediction. This noise is generated from a normal distribution with standard deviation computed from the 10% and 90% quantiles of the predictive distribution values from the BANNs for each time step. This derives from a key working assumption/approximation that the self-inferred predictive uncertainty of the BANNs is in good part due to the internal variability of the GCM climate. Our CG is trained against GCM (FAMOUS and CCSM) output

for the last deglacial interval (22 ka to present year). For predictive testing, we compare the CG predictions against GCM (FAMOUS) output for the disjoint remainder of the last glacial interval (120 ka to 22.05 ka). The CG passes a "climate Turing test", an indistinguishability test in analogy with the original Turing test for artificial intelligence. This initial validation of the Climate Generator approach justifies further development and testing for long time integration contexts such as coupled ice-sheet climate modelling over glacial cycle time-scales.

# 1 Keywords

Climate Generator, stochastic climate modelling, and emulation.





## 2   Introduction

Computational cost is a key issue for glacial cycle modelling, particularly for paleoclimate modelling. For large spatio-temporal time-scales even previous generation GCMs are prohibitively expensive, involving millions of cpu-hours for a single simulation. One practical alternative would be the statistical correction of faster simplified climate models to better approximate the

predictive quality of GCMs. This goal constitutes the central theme of this research. From a Bayesian framework, we create a statistical distribution of potential climate states, conditioned on various inputs, including the output of a fast 2D Energy Balance Model (EBM). Natural variability, or climate noise, are added to the model prediction to each time step through the addition of Gaussian noise, with noise variance extracted from the afore mentioned distribution. We focus on constructing an efficient stochastic climate representation to provide a better climate representation for glacial cycle (120 ka to present year)

modelling. Towards this goal, the small scale Weather Generator (WG) concept is implemented on a large spatio-temporal scale, and accordingly, is named a "Climate Generator (CG)".

General Circulation Models (GCMs) are an established tool for estimating the large scale evolution of the Earth's climate. They represent the physical processes occurring in the atmosphere, ocean, cryosphere and land surface and their interactions. In GCMs, core mathematical equations that are derived from physical laws (conservation of energy, conservation of mass,

conservation of momentum and the ideal gas law) are solved numerically.These models produce a three-dimensional picture of the time evolution of the state of the whole climate system. Current GCMs are too computationally expensive to run continuously over O(100 kyr) glacial cycle time scales. For example, the simplified low resolution (atmospheric part of the model has resolution $5° \times 7.5°$) FAMOUS GCM has been run for the entire last glacial cycle (LGC period, 120 ka to present year) only in a highly accelerated mode. This model can run at the rate of 250 years in a day on eight cores (Smith and Gregory, 2012). The

longest integration of a full complexity GCM to date took 2 years to complete the 22 ka to present deglacial interval (Community Climate System Model (CCSM), with a T31 resolution atmospheric component (Liu et al., 2009)). As a fast alternative, Energy Balance Models (EBMs) can integrate a whole glacial cycle in a day or less. They predict the surface temperature as a function of the Earth's energy balance with diffusive horizontal heat transports. However, in an EBM, atmospheric dynamics are not modelled and only the sea level temperature field is computed on the basis of energy conservation. Given this, the

resolution of the EBM is kept low (T11 @ 1500 km) and has no precipitation field (Hyde et al., 1990), (Tarasov and Peltier, 1997).

Coupled ice sheet and climate modelling over a full glacial cycle is an example where computational speeds currently preclude the use of GCM climate representations, especially for large ensemble-based analyses required for assessing dynamical and/or reconstructive uncertainties. Earth System Models of Intermediate Complexity (EMICSs) enable long-term climate simulations

over several thousands of years but are at the edge of applicability for a full glacial cycle. For example, LOVECLIM is a low resolution (Atmospheric component is T21) climate model. It takes about 15 days to run 10 kyr,  (Goosse et al., 2010)). Thus, there remains a need for a faster climate representation (temperature, precipitation etc.) for last glacial cycle (120 ka to present year) ice sheet modelling, especially in large ensemble contexts.



To do this, a new approach is proposed for efficient climate modelling over large spatio-temporal scales: the Climate Generator (CG). The CG uses the results of previous GCM runs to effectively improve the output of a fast simplified climate model (in this case an EBM) and thereby provide a stochastic representation of climate that runs approximately at the speed of the fast model. The Climate Generator (CG) can also be understood as a field-specific emulator for GCMs. This is because we train our

CG using GCM data to make climate predictions without the computational expense of running a full GCM. As an alternative view, the CG operates similar to aspects of downscaling tools. Downscalling tools are generally used to increase resolution in certain climate characteristics. Similarly, the CG is developed based on mainly coarse resolution climate representations and converts EBM temperature to a GCM scale. The CG produces temperature and precipitation fields using an EBM temperature field as input, similar to downscaling techniques for temperature.

Statistical Downscaling methods have been categorized, based on application technique, as regression-based methods (Multivariate Regression (MVR), Singular Value Decomposition (SVD), Canonical Correlation Analysis (CCA), Artificial Neural Network (ANN)), weather pattern based method (fuzzy classification, self-organizing map, Monte Carlo methods) and weather generators (Markov chains, stochastic models, Schoof (2013)). Regression-based methods are relatively straightforward to apply and simple to handle but have an inadequate representation of observed variance and extreme events (Wilby et al., 2004).

CCA finds spatially coherent patterns in various data fields that have the largest possible correlation, whereas SVD finds coupled spatial patterns that have maximum temporal covariance (Benestad et al., 2008). MVR optimizes the fit (minimizing the RMSE). Regression-based methods are widely used in hydrological response assessment (Chu et al., 2010). Weather-pattern based methods are often used in the analysis of extreme events. Weather Generators (WGs) replicate the statistical attributes of local climate variables rather than the observed sequences of events (Wilby et al., 2004). Regression and weather pattern based

methods have been jointly implemented via ANNs (e.g. combined principle components of multiple circulations as predictors in an ANN for winter precipitation) (Schoof, 2013). The statistical downscaling method (SDSM) is a hybrid of a regression method and weather generator (Chu et al., 2010). Statistical methods are chosen based on the nature of local predicted variables. A relatively smooth variable, such as monthly mean temperature, can be reasonably represented by regression-based methods. If the local variable is highly discontinuous in space and time, such as daily precipitation, it will require a more

complex non-linear approach (Benestad et al., 2008).

To map the relationship between large and finer scale climate aspects, Artificial Neural Networks (ANNs) are common in small spatial-temporal scale climate prediction (Schoof and Pryor, 2001). ANNs have the potential for complex non-linear input-output mapping (Dibike and Coulibaly, 2006). However, ANNs do not have associated uncertainty estimates, and over-fitting is a hazard. To minimize over-fitting and to find an optimum network, ANNs rely on a cross-validation test. Cross-validation

does not use training data efficiently as it requires disjoint data sets for testing and parameter estimation. Bayesian Artificial Neural Networks (BANNs) generate uncertainty estimates and avoid the need for cross-validation. In BANNs, an assumed prior distribution of parameters (weight and biases) is used to specify the probabilistic relationship between inputs and outputs. The prior distribution is updated to a posterior distribution by a likelihood function through Bayes theorem. The predictive distribution of the network output is acquired by integration over the posterior distribution of weights. BANNs are used in

different applications e.g., to create weather generators (Hauser and Demirov, 2013) , for model calibration (Hauser et al.,



2012), and for short time scale climate prediction (Maiti et al., 2013), (Luo et al., 2013). Our CG uses BANNs to estimate a posterior distribution for climate prediction/retrodiction conditioned on various inputs including the output of an EBM.

Our primary aim is to create a fast, efficient stochastic climate representation for glacial cycle scale modelling. To date, most glacial cycle ice sheet simulations use a glacial index (for example Tarasov et al. (2012)) in a combination with an LGM

timeslice of GCM output for their climate forcing. Unless full energy-balanced surface mass-balance is being computed, ice-sheet models for glacial cycle contexts do not require accurate year to year prediction in their climate forcing. The dynamical memory of ice sheets filters interannual fluctuations. As such, 30 to 50 year scale monthly mean climatologies with statistics for shorter scale temperature variability are a reasonable trade-off for enabling computation over such timescales. Given this context, we invoke the concept of a Climate Generator (CG) for improving the output of fast EMICS (in this case a geographi-

cally resolving 2D EBM) for 100 kyr scale contexts. The CG is a large spatio-temporal scale dynamic representation of climate based on a regressed relationship between the EBM (and various other predictor variables) and relevant GCM outputs.

## 2.1 Climate Generator (CG)

### 2.1.1 Reasoning behind the name of CG

Weather Generators (WGs) are computationally useful statistical tools that can be used to generate realistic and rapid daily

sequences of atmospheric variables, such as temperature and precipitation, on a small scale. WGs are a means of generating a random time series of 'weather' that replicates observed statistics. It can be used to investigate small-scale climate impacts and to compute numerous random realizations quickly. Moreover, WG outputs are set to the observed distributional properties, primarily on the daily or sub-daily scale (Ailliot et al., 2015). In this research, we implement the Weather Generator concept on a large spatial-temporal scale and subsequently propose the term Climate Generator (CG).

### 2.1.2 CG predictors

The climate of a geographic location has various strong dependencies, such as latitude, earth-sun relationships, proximity to large bodies of water, atmospheric and oceanic circulation, topography and local features. The climatological temperature field is relatively smooth, depending most strongly on latitude, surface elevation and continental position. However, the climatological precipitation field is not smooth and has strong longitudinal and non-local dependence. Like a WG, our CG generates

a synthetic climatology conditioned on various inputs. In probabilistic terminology, the CG provides a posterior distribution for climate prediction conditioned on the given predictors. Moreover, our CG predicts temperature and precipitation fields jointly, so predictands are correlated with each other. The CG presented herein predicts monthly mean surface temperature and precipitation fields by considering the above characteristic of climate through predictor variables: latitude, longitude, monthly mean sea surface temperature field from a fast low-resolution Energy Balance Model (EBM), surface elevation, ice mask,

atmospheric concentrations of carbon dioxide and methane, and orbital forcing. Some predictor variables (latitude, longitude, carbon dioxide etc.) are already taken into account by the EBM, but perhaps to an inadequate extent. We therefore explicitly include those predictor variables to create our CG and then test whether their inclusion is required via Automatic Relevance

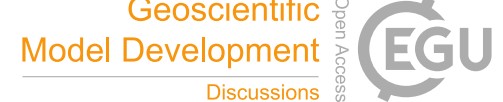



Determination (ARD) Neal (2012). Latitude and longitude are included in our study because they have direct effects on climate prediction. We select surface elevation as a predictor due to strong vertical temperature gradients that vary laterally. Due to its high albedo, the presence of ice is also included as a predictor. Carbon dioxide and methane data are considered as a predictor given their relatively significant radiative forcing variations over a glacial cycle. Finally, we include monthly mean sea surface
EBM temperature in our predictor sets, since the EBM can estimate the sea level monthly mean surface temperature very efficiently. As an attempt to capture non-local effects, the first two EOFs of surface elevation and the area of ice were tested in our CG as predictors but did not yield any significant improvement in CG predictions.

### 2.1.3  Climate Generator and reality

Computational tools, which are used in climate variability representation, may have many sources of uncertainty. For example,
GCMs uncertainties are classified into initial conditions, boundary conditions, parametric and model structure. For the hypothetical case of accurate initial and boundary conditions, and accurate model parameter values, the model structural uncertainty is then isolated:

$$\text{Reality}(t) = \text{GCMs}(t) + \alpha, \tag{1}$$

where $\alpha$ is the structural error. The CG emulates GCMs, so all GCMs uncertainties listed above propagate into the CG. The
BANN also estimates its own regression uncertainty. We make the assumption that this regression uncertainty is largely due to smaller spatio-temporal scale dynamics and non-local couplings within the GCM and thereby consider it as climate noise. To capture GCM variance, this predictive uncertainty of the BANN is used to specify the variance of uncorrelated Gaussian noise that is added to each CG prediction. This uncorrelated aspect of the injected "climate noise" is a further source of error. We did test the inclusion of the first two surface elevation EOFs in the CG input set, but no significant improvements arose. This CG
does not account for the structural error of the GCM (which would on it's own be an extensive investigation).

### 2.1.4  Climate Turing Test

In the research of artificial intelligence, the Turing Test is performed to determine if a machine's ability to exhibit intelligent behaviour is indistinguishable from that of a human. The machine will pass the test if it responds to inquires in a manner that is indistinguishable from how a human could potentially respond. In this project, the Turing test concept is implemented as a
"Climate Turing Test (CTT)", to ascertain if the pre glacial maximum behaviour (120 ka to 22.05 ka) of climate predicted by our CG is indistinguishable from the FAMOUS climate model. To execute the Climate Turing test, the difference between CCSM and FAMOUS is taken as a reference uncertainty to compare the difference between the CG and FAMOUS. The comparisons are made based on the mean correlation (over space and time), mean deviation, Root Mean Square Error (RMSE), map plots and projections of Empirical Orthogonal Function (EOF) of our CG predictions with FAMOUS climatological simulated fields
(monthly mean temperature and precipitation). To pass the Climate Turing test, simulated fields should have relatively high correlation (i.e. with respect to GCM target field), small RMSE, close patterns and a reasonable capacity to capture natural variability compared to the reference uncertainty. The goal here is to think beyond optimizing a regression to a given metric(s),

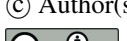



without considering what behaviour obtains this "high score". Rather, use a wide variety of metrics designed to assess if the emulator really "acts like" the climate simulation it is being used as a stand in for.

## 3 Test study region and data

The test study region of interest is North America (including Greenland; more specifically, longitude 188E : 355E and latitude 34N : 86N). This combines a continent that experienced past glaciation but has no significant present-day ice cover with a region that has had continued ice cover to present. For methodological validation, we use the FAMOUS (low-resolution GCM) climate model data sets (Smith and Gregory, 2012). The atmospheric portion of the model has a resolution of $5° × 7.5°$). We take the 1.5 meter air temperature and precipitation fields, and train our CG (CGfamous) over a time period of 22 ka to present year. The FAMOUS model was run with an accelerated mode (factor 10) for the full Last Glacial Cycle (LGC) (120 ka to present). This allows us to divide the full-time interval into two parts: the training interval of 22 ka to the present year, and a test interval of 120 ka to 22.05 ka. Fifty year climatologies (5 consecutive years for FAMOUS, given the factor 10 accretion) are considered. For this initial test of concept, calibration and validation of the CG is implemented for the coldest month (February) and the warmest month (August). Each month has a total dataset of 2400 time steps for FAMOUS (120 ka to present year), with 440 time steps for training (22 ka to present), and each time step has 231 (21x11) gridcells (FAMOUS). The CG is trained to predict values for each grid cell. For model validation, we compare the CGfamous output directly to that of FAMOUS over the test interval. For further evaluation, we compare CGfamous values with those produced by the EBM (T11 @1500 km) and an additional climate model, CCSM3 (Monthly means, T31) climate model (Liu et al., 2009), over the test period. For the comparisons, EBM output (Deblonde et al., 1992) sea level temperature is adjusted to surface temperature with a lapse rate of 6.5 K/km. For the purpose of our Turing test, the difference between the results of the two GCMs (FAMOUS and CCSM3) are taken as a minimum structural error estimate for FAMOUS.

We also train the CG to the output of a much more advanced GCM, CCSM3 (CGccsm). We use CCSM3 monthly mean radiative surface temperature and precipitation, with 50 year climatologies estimated using 5 samples spaced 10 years apart. This creates a training data set with 440 time steps for 22 ka to present year and and 602 (43x14) grid cells. The goal here is to test our CG creation procedure under the demands posed by a higher resolution climate model than FAMOUS. However, since the CCSM3 experiment only runs from 22 ka to present, there is no test period available for validation.

## 4 Methods

The CG uses Bayesian Artificial Neural Networks (BANNs) for estimating an evolving climate state as a function of various inputs. The BANNs also estimate predictive uncertainty which (in an arguable leap) we take to represent the shorter scale un-resolved variability in theclimate. BANNs are effectively a set of artificial neural networks with individual parameters from a posterior probability distribution derived from training the network against observed input-output sets. The CG estimates target values based on the mean from the resulting set of networks, and its squared error.





## 4.1 BANN design and training

We design our ANN architecture by using the software for flexible Bayesian modelling package (freely available) at http://www.cs.utoronto.ca/radford/fbm.software.html. Different architectures (number of hidden layers, node size, connection of inputs, hidden units and outputs) and different predictors are first tested. Architectures are selected according to the predictive skill on the test data. Step size and prior specification are then adjusted to improve prediction capability. To choose an appropriate predictor set, we tested the following predictor variables: latitude, longitude, EBM sea level temperature (Deblonde et al. (1992)), carbon dioxide (Lüthi et al. (2008)), methane (Loulergue et al. (2008)), surface elevation (Smith and Gregory, 2012), surface type (ice) (Smith and Gregory, 2012), orbital forcing for various seasons and latitudes, temperature and precipitation ((Smith and Gregory, 2012) and (Liu et al., 2009)), melt water flux (five different locations; (Smith and Gregory, 2012)),first two EOFs of ice surface elevation, and ice area. The outputs are monthly mean surface temperature and precipitation. Various network architectures and different combinations of predictor variables were tested. ARD was used to identify which predictors provide meaningful weight in the distribution value. Various combinations of the input set and network architectures were also evaluated against the test interval subset of the GCM output (as detailed below). The resultant optimal input set is comprised of: latitude, longitude, surface elevation, ice, carbon dioxide, methane, orbital forcing (June/July/August mean solar insolation at 60N), and EBM sea level temperature. The best fitting BANN architecture has two hidden layers, both using $tanh(x)$ as the neuron transfer function, as detailed in Figure 1.

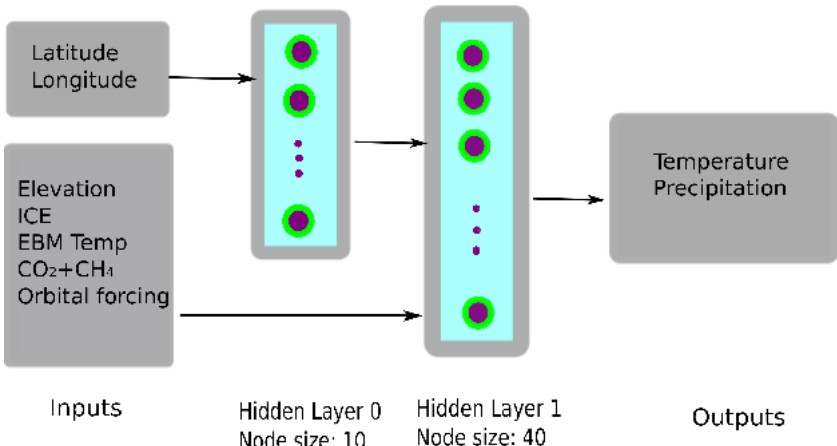

**Figure 1.** Final ANN Diagram (Architecture type I) used in CGfamous and CGccsm prediction

## 4.2 BANN Implementation

Bayesian Artificial Neural Networks (BANNs) estimate a probability distribution and are derived by training the network against available data using Bayesian inference. Markov Chain Monte Carlo (MCMC) sampling is used for selecting the distribution parameters for the networks. The step by step procedure for BANN's implementation follows Neal (2012):



- Process predictor and predictand data sets.

- Define architecture and specify model to have real valued targets.

- Define a prior distribution for parameters (weight and biases). The prior is specified hierarchically for a group of parameters with a three-level approach as follows. Each prior is sampled from a Gaussian distribution with zero mean and some precision (inverse standard deviation). The precision for a "sub-group" of parameteris in turn selected from a Gamma distribution with a selected shape parameter and with a mean given by a hyperparameter. Finally, this hyperparameter is sampled from a Gamma distribution with a specified mean and with another assigned shape parameter. Mean precision at the top level is assigned as 0.1 (width) and the shape parameters of the Gamma distribution are assigned a value of 2. Priors for input to hidden layer, connections between hidden layers and hidden layers to outputs are automatically re-scaled based on the number of hidden units and the prior of the output bias is specified as a Gaussian prior with mean zero and standard deviation 10.

- A noise model is fitted to estimate network parameters. To fit a noise model each prediction for targets is considered to be a sample from a Gaussian distribution. The noise levels specification follows the same three level approach used for specification of the priors. The targets are modelled as the network outputs plus Gaussian noise.

- Specify data for training and testing.

- Initialize the network with hyperparameters set to (say) 0.3 and all parameters set to zero. Markov chain operations are defined, where each iteration consists of over fifty repetitions of the following steps: Gibbs sampling for the noise level, hybrid Monte Carlo sampling using a fixed trajectory and step size adjustment factor. In this stage, the hyperparameters are not updated and remain fixed at a value 0.3.

- A single iteration of the above process is representative of one step in Markov chain simulation. The rejection rate is examined after a number of (say 50) hybrid Monte Carlo updates. If the rejection rate is high (say, over 40%), the Markov chain simulation is repeated with a smaller step size adjustment factor.

- A network is stored in the log file containing the parameters and hyperparameters values. Markov chain sampling is repeated and overrides the previous set. Each iteration consists of say five repetitions of the following steps: Gibbs sampling for both the hyperparameters and the noise level followed by hybrid Monte Carlo sampling as above. (A long trajectory length is useful for the sampling phase).

- By looking at the hyperparameters values and quantities such as the squared error on the training set, we can get an idea of when the simulation has reached equilibrium or not. After that, we can start prediction.

- Generate predictions (mean, 10% and 90% quantiles) for the test cases from the resultant distribution of networks. By using different initial seeds, ensembles of several networks are generated by sub-sampling from the later segments of the Markov Chains.





The detailed implementation procedure is given in Neal (2012).

### 4.3 Adding noise

Gaussian noise is added in our CG prediction to account for (at least in part) seasonal to decadal climatic variability not captured by the EBM. This natural variability (noise) is physically correlated across space and time. However, given the context (coupling with ice sheet models for glacial cycle timescales) and for computational simplicity, the CG noise injection uses uncorrelated random sampling. Ice sheet thermodynamic response to climate is smoothed to centennial or longer timescales. Surface mass-balance response for the given grid scales will be sensitive to the variance of temperature but not to spatial correlations nor much to temporal correlations. Correlations between temperature and precipitation could have significant impact, especially during the potential melt season. The August residual (CG without noise - FAMOUS) correlation map between the temperature and precipitation fields shows magnitudes of mostly less 0.3 (Figure 4 in the supplement) and are therefore relatively small (this does not, however, rule out significant non-linear relationships). The random noise is added to each time step of our CG predictions by generating a random sample from Gaussian distribution with $\sim N(\mu = 0, \sigma)$, where $\mu$ = Mean and $\sigma$ = Standard deviation (80% confidence interval scale). The standard deviation is computed through the BANNs predicted 10th percentile and 90th percentile of the predictive distribution of a single guess for each case. Standard deviation is computed from the following Equation:

$$\sigma = \frac{(X_{90\%} - (X_{90\%} + X_{10\%})/2)}{Z_{90\%}} \tag{2}$$

where $Z_{90\%}$ = 1.28 (Z values or score), calculated from the statistical table. The values of $\sigma$, defined in the Equation 2 had space and time dependence. This assumption that BANN predictive uncertainty can provide an approximate estimate for the unresolved climatic variability is tested in part below.

## 5  Implementation Results

This research implements the CTT concept to measure the prediction capability of our CG. The CTT determines whether or not CGfamous is capable of predictions of the same style as the FAMOUS climate model. To implement the CTT concept, a direct comparison was done between CGfamous and FAMOUS Climate model outputs over the test interval (120 ka to 22.05 ka) using characteristics such as Root Mean Square Error (RMSE), visual patterns in map plots and Empirical Orthogonal Function (EOF) projection to measure the difference in outputs between CGfamous and FAMOUS relative to that between CCSM and FAMOUS. Here we give some example comparisons for different architectures and predictor sets.

### 5.1  Selection of BANN architecture

More than 100 different BANN architectures (different number of hidden layers, different connections and node sizes) with different combinations of predictor sets were tested. To convey the sensitivity to architecture and predictor set, we present





results for three basic architectures (Figure 1 and Figure 2) in combination with various predictor sets (Table 1). Architecture type I (Figure 1) gave the best overall fit of CGfamous to FAMOUS over the test interval.

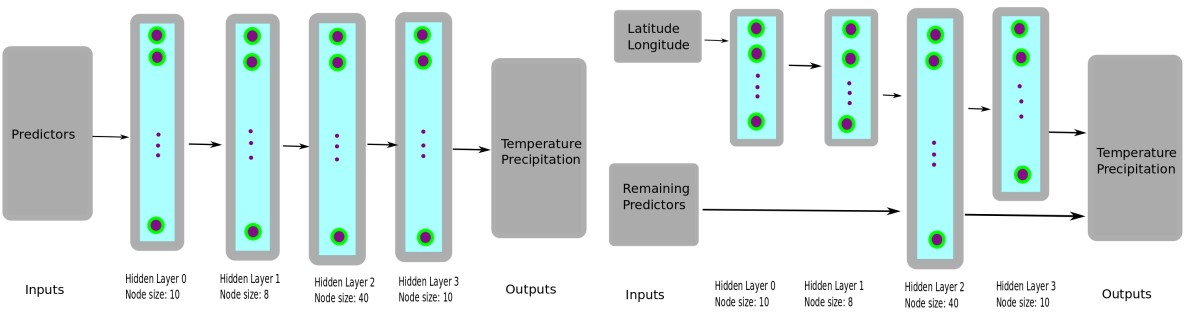

(a) Architecture type II                     (b) Architecture type III

**Figure 2.** Diagrams of some of the different ANN architectures tested during development

**Table 1.** Predictor sets for displayed CG results

| |
| --- |
| Combination 1: PS1 = Latitude, Longitude, surface elevation, ice, EBM temp |
| Combination 2: PS2 = PS1, $CO_2 + CH_4$, orbital forcing, melt water flux |
| Combination 3: PS3 = PS1, carbon dioxide, methane, orbital forcing |
| Combination 4: PS4 = PS3, EOF-1 and EOF-2 of ice data, ice volume |

**Table 2.** Different models based on architecture and inputs

| Model Name | Predictors | Architecture type |
| --- | --- | --- |
| A | PS3 | II |
| B | PS1 | I |
| C | PS3 | III |
| D | PS2 | I |
| E | PS4 | I |
| F | PS3 | I |

Six model names are assigned in Table 2 based on different predictor sets and architecture (Table 1).





**Table 3.** MEAN DIFFERENCE (MD) and RMSE relative to the FAMOUS (except for last entry) over the full grid area (train period). The model $M$ is the same as model $F$ but trained with CCSM climate model data.

| Model | TEMP (DEG C) | | | | PREC (cm/month) | | | |
| | FEB | | AUG | | FEB | | AUG | |
| | MD | RMSE | MD | RMSE | MD | RMSE | MD | RMSE |
|---|---|---|---|---|---|---|---|---|
| A | 3.91 | 9.07 | 4.09 | 5.7 | 0 | 3 | 1 | 3 |
| B | 2.85 | 10.19 | -2.93 | 9.9 | 1 | 4 | 2 | 5 |
| C | 2.43 | 7.94 | 5.45 | 5.36 | 1 | 3 | 1 | 4 |
| D | -3.3 | 4.13 | -1.73 | 2.67 | 0 | 2 | 0 | 2 |
| E | -2.35 | 5.77 | -1.05 | 4.6 | 0 | 2 | 1 | 3 |
| F | -0.45 | 3.92 | -1.23 | 2.52 | 0.5 | 3 | 0 | 3.1 |
| M (1) | 5.87 | 10.51 | -4.4 | 4.29 | 0.02 | 3.08 | 0 | 3.06 |
| CCSM (2) | 5.86 | 10.99 | -4.4 | 4.08 | 0.5 | 4.1 | -1.3 | 4.1 |
| EBM | 12.46 | 11.0 | -1.52 | 4.77 | | | | |
| (1) versus (2) | -0.02 | 4.14 | 0.01 | 2.67 | 0.5 | 4.1 | -1.3 | 4.1 |

These models are compared with FAMOUS model outputs over the training interval in Table 3 and over the test interval in Table 4.

**Table 4.** MEAN DIFFERENCE (MD) and RMSE relative to the FAMOUS (Test period) over the full grid area

| Model | TEMP (DEG C) | | | | PREC (cm/month) | | | |
| | FEB | | AUG | | FEB | | AUG | |
| | MD | RMSE | MD | RMSE | MD | RMSE | MD | RMSE |
|---|---|---|---|---|---|---|---|---|
| A | 4.45 | 8.08 | 3.93 | 9.1 | -0.16 | 2.5 | 1.07 | 4 |
| B | 3.35 | 9.11 | -3.09 | 8.5 | 0.83 | 3 | 2.07 | 5 |
| C | 3.2 | 6.49 | 5.29 | 12.5 | -0.18 | 5 | 0.07 | 3 |
| D | -2.75 | 5.74 | -1.29 | 5.22 | -0.16 | 2.3 | -0.93 | 3 |
| E | -1.75 | 5.38 | -1.2 | 4.68 | -0.15 | 2.3 | 0.07 | 3.4 |
| F | -2.45 | 5.49 | -1.45 | 2.98 | -0.15 | 1.9 | 0.07 | 2.8 |
| M | -4.26 | 11.27 | -6.64 | 5.36 | 0 | 3 | 0.03 | 3.3 |
| EBM | 10.82 | 10.45 | -5.3 | 4.76 | | | | |

The positions of each letter appearing in Figure 3 and 4 quantifies how closely that model's simulated temperature and precipitation pattern match the FAMOUS climate model outputs and gives a graphical summary of comparisons of the RMSE and standard deviation. RMSE is computed from the differences between different CG predictions with FAMOUS climate model



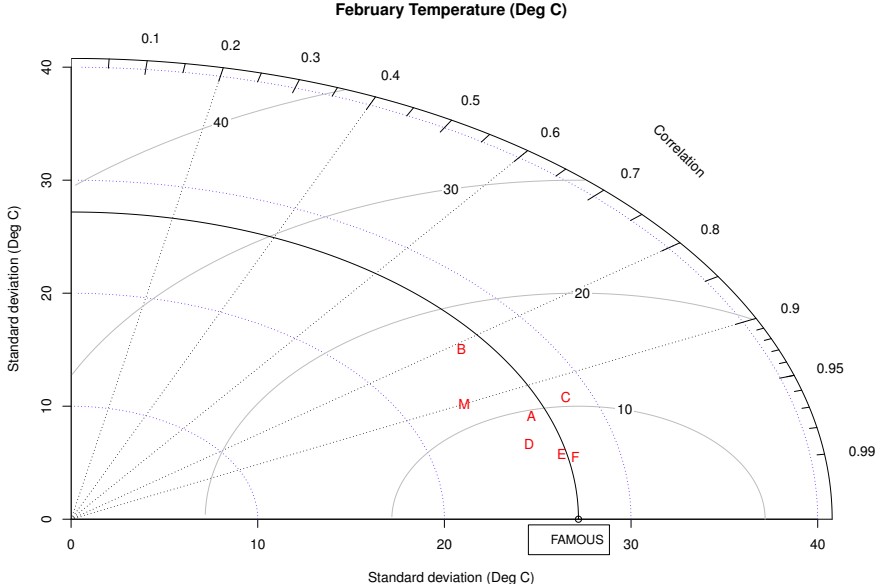

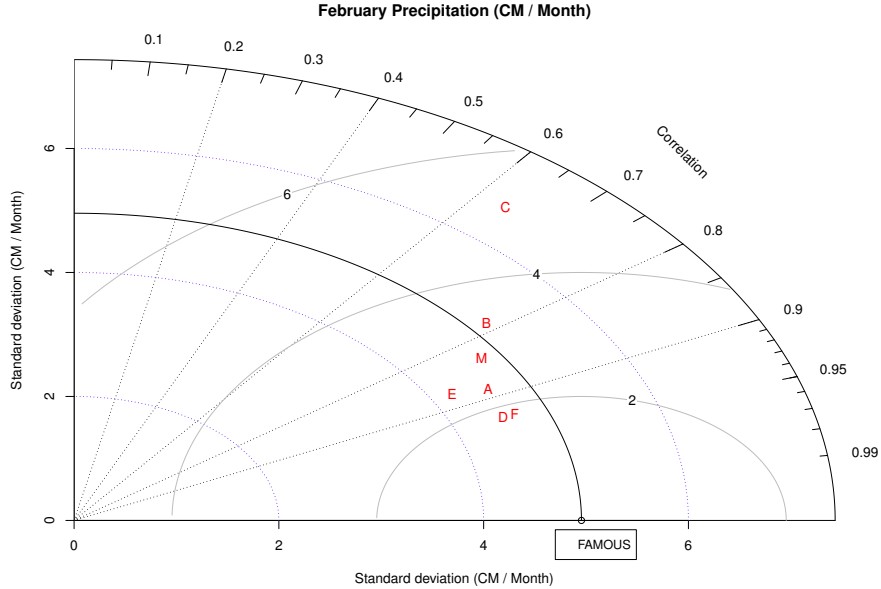

**Figure 3.** Taylor diagram displaying a statistical comparison with FAMOUS (test part) of six different CG estimates. Pattern similarities are quantified based on their correlation and centered root mean square difference between CG and FAMOUS, and standard deviation with respect to the mean of the corresponding field. Contour grey lines indicate the root mean square (RMS) values. The model $M$ is the same as model $F$ but trained with CCSM climate model data.



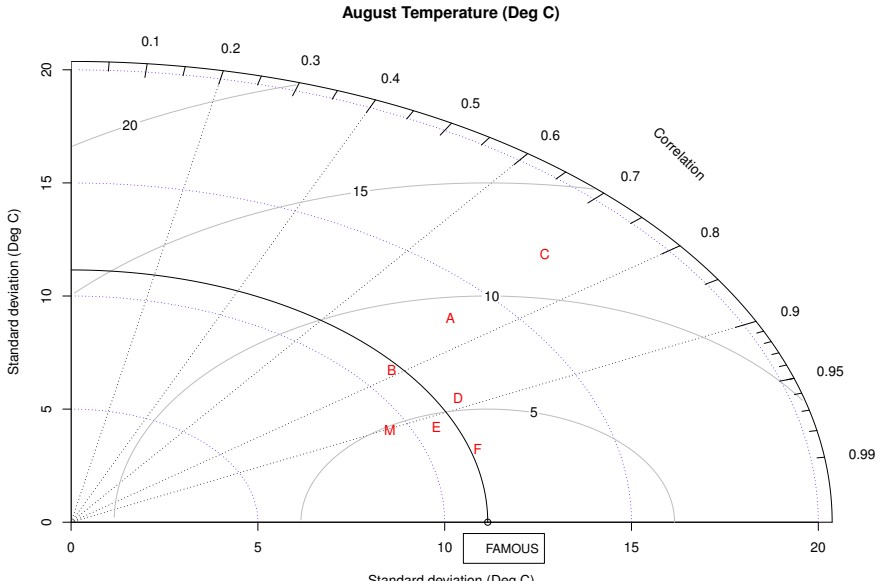

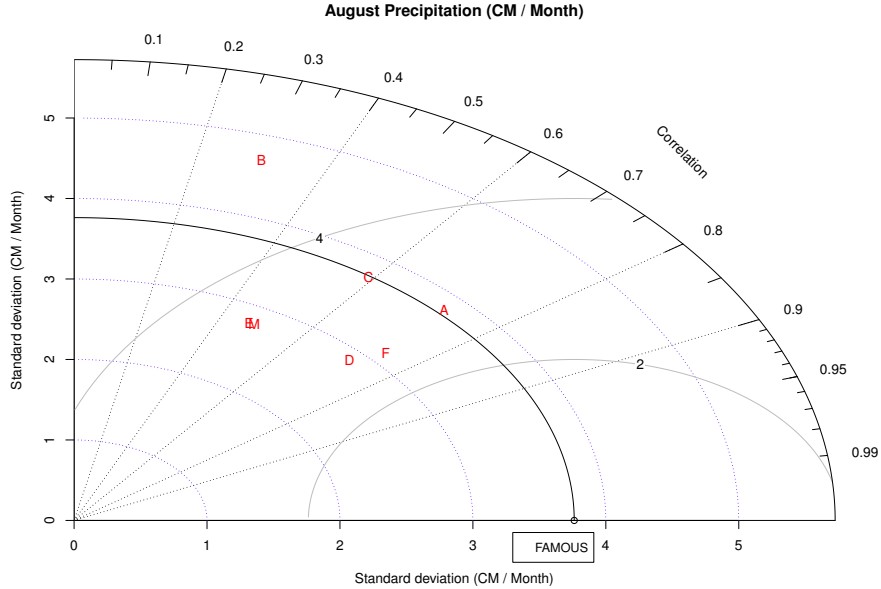

**Figure 4.** Taylor diagram displaying a statistical comparison with FAMOUS (test part) of six different CG estimates. Pattern similarities are quantified based on their correlation and centered root mean square difference between CG and FAMOUS, and standard deviation with respect to the mean of the corresponding field. Contours grey line indicates the root mean square (RMS) values. The model $M$ is the same as model $F$ but trained with CCSM climate model data.





outputs. The Standard deviation of FAMOUS is indicated by the black contour line in Taylor diagrams. Our main selection criterion is the minimization of RMSE. Beside this, mean deviation (general bias) comp arison on Table (4) and standard deviation and correlation from Figure 3 and Figure 4 allow us to see how comparable our models prediction are with FAMOUS. The central RMSE is about $2.98\,^{\circ}$C (August temperature) and $5.49\,^{\circ}$C (February temperature) for model $F$, which are the

lowest compared to all other models in Table 4. In the case of precipitation, the RMSE of model $F$ is about 1.9 cm/month (February) and 2.8 cm/month (August), which also are the least compared to all other models. The model $F$ has strong mean correlation (over space and time) with FAMOUS at about 0.98 for February temperature and 0.92 for February precipitation in Figure (3 and 4). The model $F$ has the best fit compared to other models listed in Table 4. As is evident in Tables 3 and 4 and figures (3 and 4), there is significant sensitivity to network architecture and predictor set with a factor 4 range in temperature

RMSE and factor 2 range in precipitation RMSE for the example combinations. The sensitivity to predictor sets is quantified in more detail through ARD.

To choose predictor sets, an ARD analysis has been done. The ARD test helped us to choose which inputs are relevant for the outputs and is determined based on the hyperparameters values which control the standard deviation for weights and biases in different groups. In Neal (2012), these hyperparameters values are referred to as "sigma" values. The significance of each

input is non-linearly proportional to the Sigma value in Table 5.

**Table 5.** ARD analysis for the Model $F$ (CGfamous)

| Predictors | Sigma value |
|---|---|
| Latitude | 1.30 |
| Longitude | 0.67 |
| Elevation | 2.85 |
| Ice | 2.79 |
| CO2+CH4 | 1.18 |
| Orbital forcing | 3.42 |
| EBM temp | 4.46 |

For the case of Model $F$, EBM temperature is the most significant input. However, even though the EBM computes orbital forcing and accounts for greenhouse gases, orbital forcing is still the next most significant input. Longitude is the least important, even though temperature and precipitation have in reality high dependence on longitude given atmospheric circulation dependencies. The EBM does have a slab ocean with thermodynamic sea ice, and this result suggests that continentality effects

might be reasonably captured by the Model EBM. Further comparisons were carried out with field plots and EOF projections for all simulated fields. Model $F$ predictions have the least RMSE (compared to FAMOUS) among all tested models. It has the highest positive correlation and comparable mean difference (with FAMOUS). Therefore the model $F$ has been chosen and renamed CGfamous. The prediction capability of our model CGfamous (trained with FAMOUS model data) and CGccsm (trained with CCSM data) are tested by implementing CG over two other continents, namely Antarctica and Eurasia.





## 5.2   Model Comparison

Weather Generator performance can be tested against new observations, but such opportunities are limited in our case. The CG is evaluated based on computing statistics (RMSE, Mean deviation, correlation), the goodness of fit (including noise levels) and qualitative consideration against FAMOUS climate model outputs (test interval). The best and worse fits are identified

5   between CG prediction and FAMOUS outputs on specific regions or latitude bands, winter versus summer and full grid area versus ice region. The comparison over the ice region gives the opportunity to check the prediction capability of our CG in the context of glacial cycle modelling. Criteria such as space-time scale appropriateness, patterns, and climate noise variability at shorter time scales are introduced to measure variance. The difference between CCSM and FAMOUS over the training interval is taken as a minimum value of model uncertainty and thereby our reference misfit bound for the climate Turing Test.

**Table 6.** Mean Difference (MD), Correlation (COR) and RMSE relative to FAMOUS (except for last entry) over the full grid area (Train period)

| | Model | TEMP (Deg C) | | | | | | PREC (cm/month) | | | | | |
| | | FEB | | | AUG | | | FEB | | | AUG | | |
| | | MD | RMSE | COR | MD | RMSE | COR | MD | RMSE | COR | MD | RMSE | COR |
|---|---|---|---|---|---|---|---|---|---|---|---|---|---|
| North America | EBM | 12.46 | 11.0 | 0.86 | -1.52 | 4.77 | 0.80 | | | | | | |
| | CGfamous | -0.45 | 3.92 | 0.92 | -1.23 | 2.52 | 0.90 | 0.5 | 3 | 0.85 | 0.0 | 3.1 | 0.80 |
| | CGccsm(1) | 5.87 | 10.51 | 0.88 | -4.4 | 4.29 | 0.89 | 0.02 | 3.08 | 0.87 | 0.0 | 3.06 | 0.85 |
| | CCSM(2) | 5.86 | 10.99 | 0.99 | -4.4 | 4.08 | 0.99 | 0.5 | 4.1 | .77 | -1.3 | 4.1 | 0.82 |
| | (1) versus (2) | -0.02 | 4.14 | 0.99 | 0.01 | 2.67 | 1.0 | 0.5 | 4.1 | 0.89 | -1.3 | 4.1 | 0.95 |
| EurAsia | EBM | 10.07 | 11.17 | 0.44 | -4.54 | 4.38 | 0.75 | | | | | | |
| | CGfamous | 0.64 | 5.57 | 0.40 | 0.22 | 3.07 | 0.59 | 0.32 | 3.2 | 0.02 | 2.18 | 5.09 | -0.04 |
| | CGccsm(1) | 2.26 | 7.13 | 0.45 | -7.12 | 4.76 | 0.76 | 0.45 | 4.2 | 0.02 | -1.86 | 5.05 | 0.06 |
| | CCSM(2) | 2.23 | 6.92 | 0.46 | -7.10 | 4.23 | 0.71 | 0.33 | 4.21 | 0.13 | -1.9 | 5.25 | 0.11 |
| | (1) vs (2) | 0.03 | 4.23 | 0.82 | -0.02 | 3.34 | 0.88 | 0.12 | 2.02 | 0.16 | 0.07 | 2.05 | 0.24 |
| Antarctica | EBM | -0.79 | 3.78 | 0.48 | 18.33 | 9.73 | 0.47 | | | | | | |
| | CGfamous | -0.041 | 2.37 | 0.22 | 0.639 | 4.75 | 0.43 | 0.51 | 2.38 | -0.01 | 0.49 | 1.87 | -0.01 |
| | CGccsm(1) | -4.99 | 3.83 | 0.45 | 7.62 | 7.81 | 0.48 | -0.95 | 2.40 | 0.06 | 0.35 | 1.81 | 0.12 |
| | CCSM(2) | -4.99 | 4.19 | 0.48 | 7.91 | 8.09 | 0.46 | -2.75 | 2.90 | 0.09 | -1.84 | 2.62 | 0.16 |
| | (1) vs (2) | -0.001 | 2.29 | 0.95 | 0.002 | 3.02 | 0.96 | 1.81 | 1.09 | 0.46 | 2.19 | 1.88 | 0.62 |

10   For both the test and training over North America, the RMSE of CGfamous simulated temperature field (about FAMOUS) is about 50% less than the RMSE of CCSM and the EBM model (about FAMOUS) over the full grid area. For precipitation over the test interval, the RMSE of CGfamous (about FAMOUS) is approximately 42% less than the RMSE of CCSM (CCSM versus FAMOUS). The CG simulated August temperature has a better fit with FAMOUS (less RMSE) compared to that of



February. The CG appears to work better in the ice region (approximately 54% or more less RMSE compare to the full grid area). In addition to these quantitative statistics, CGfamous also has improved the geographic pattern of misfits (CGfamous -FAMOUS versus FAMOUS - CCSM and FAMOUS - EBM).

**Table 7.** Mean Difference (MD), Correlation (COR) and RMSE relative to FAMOUS (except for last entry) over the ice region (Train period)

| | Model | TEMP (Deg C) | | | | | | PREC (cm/month) | | | | | |
| | | FEB | | | AUG | | | FEB | | | AUG | | |
| | | MD | RMSE | COR | MD | RMSE | COR | MD | RMSE | COR | MD | RMSE | COR |
|---|---|---|---|---|---|---|---|---|---|---|---|---|---|
| North America | EBM | 1.58 | 4.41 | 0.99 | -0.38 | 1.76 | 0.98 | | | | | | |
| | CGfamous | 0.11 | 1.33 | 1.0 | 0.02 | 0.53 | 1.0 | 0.05 | 0.41 | 0.87 | 0.01 | 0.92 | 0.95 |
| | CGccsm(1) | 1.06 | 3.40 | 0.99 | -1.23 | 1.24 | 0.99 | 0.06 | 0.38 | 0.86 | 0.01 | 0.89 | 0.96 |
| | CCSM(2) | 1.06 | 3.58 | 0.99 | -0.27 | 1.19 | 0.99 | -0.01 | 0.39 | .87 | -0.16 | 0.78 | 0.96 |
| | (1) versus (2) | 0.0 | 0.92 | 0.99 | -0.02 | 0.51 | 1 | 0.02 | 0.30 | .86 | 0.16 | 0.54 | 0.96 |
| EurAsia | EBM | 0.18 | 0.99 | 0.98 | -.015 | 0.16 | 0.90 | | | | | | |
| | CGfamous | 0.12 | 0.68 | 0.98 | 0.03 | 0.18 | 0.19 | 0.02 | 0.14 | 0.62 | 0.06 | 0.44 | 0.82 |
| | CGccsm(1) | 0.08 | 0.51 | 0.97 | -.14 | 0.83 | 0.90 | 0.009 | 0.01 | 0.49 | 0.02 | 0.21 | 0.77 |
| | CCSM(2) | 0.08 | 0.65 | 0.98 | -0.11 | 0.53 | 0.91 | 0.005 | 0.06 | 0.72 | -0.02 | 0.18 | 0.84 |
| | (1) versus (2) | 0 | 0.41 | 0.99 | -0.03 | 0.45 | 0.99 | 0.004 | 0.09 | 0.63 | 0 | 0.1 | 0.83 |
| Antarctica | EBM | 0.16 | 3.43 | 0.41 | 14.6 | 11.47 | 0.47 | | | | | | |
| | CGfamous | 0.11 | 2.20 | 0.21 | 0.28 | 3.34 | 0.43 | 0.31 | 1.45 | -0.01 | 0.43 | 1.19 | -0.003 |
| | CGccsm(1) | -4.58 | 4.08 | 0.40 | 5.46 | 6.38 | 0.48 | -0.38 | 1.55 | 0.05 | 0.29 | 1.11 | 0.12 |
| | CCSM(2) | -3.72 | 4.37 | 0.43 | 5.52 | 6.72 | 0.47 | -1.131 | 2.17 | 0.10 | -0.63 | 1.47 | 0.16 |
| | (1) versus (2) | 0.001 | 2.17 | 0.97 | 0.061 | 2.71 | 0.97 | 0.93 | 1.07 | 0.46 | 0.92 | 1.41 | 0.56 |

The CGfamous simulated temperature field (both months) has a clear cold bias (in test part) compared to FAMOUS. But
5   CGccsm has a sharp and warm bias (February) in addition to a clear cold bias (August) compared to the FAMOUS test map. The CG simulated temperature field has captured approximately 70% temporal variance (based on the leading two EOFs) of that of FAMOUS. Precipitation is a challenge for all models, and the CGfamous precipitation field captures about only 40% (based on leading two EOFs) of the temporal variance compared to FAMOUS. For context, the CCSM precipitation field is also far from that of FAMOUS. For both months over the training period, the CGfamous (network $F$) temperature RMSE
10  relative to FAMOUS is less than half of our structural uncertainty reference (i.e. CCSM - FAMOUS) and the corresponding precipitation RMSE is about 33% smaller (TABLE 7). The temperature RSME not unexpectedly increases over the predictive test region, but critically these values are still about half of the CCSM - FAMOUS reference RMSE. CGfamous temperature is highly correlated with FAMOUS (0.96 or higher) over the test interval with a temperature bias (MD) that is 2 degree (about 40%) lower than the CCSM bias over the test interval (Table 9). As structural uncertainty is larger than the difference between

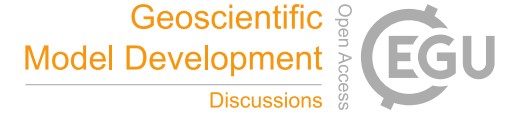

CCSM and FAMOUS, CGfamous temperature largely passes the Climate Turing Test for these metrics. The approximate factor 2 improvement in RMSE of CGfamous versus that of the EBM validates the CG role of statistical improvement of the fast simplified models over North America. Similar improvements are evident for Antartica and Eurasia.

**Table 8.** Mean Difference (MD), Correlation (COR) and RMSE relative to FAMOUS (except for last entry) over the full grid area (Test period)

| | Model | TEMP (Deg C) | | | | | | PREC (cm/month) | | | | | |
| | | FEB | | | AUG | | | FEB | | | AUG | | |
| | | MD | RMSE | COR | MD | RMSE | COR | MD | RMSE | COR | MD | RMSE | COR |
| N. America | EBM | 10.82 | 10.45 | 0.89 | -5.3 | 4.76 | 0.91 | | | | | | |
| | CGfamous | -2.45 | 5.49 | 0.97 | -1.45 | 2.98 | 0.96 | -0.15 | 1.9 | 0.92 | 0.07 | 2.8 | 0.77 |
| | CGccsm | -4.26 | 11.27 | 0.90 | -6.64 | 5.36 | 0.90 | 0.0 | 0.82 | 0.03 | 3.3 | 0.5 | 0.88 |
| E.Asia | EBM | 9.5 | 11.22 | 0.45 | -6.52 | 3.95 | 0.60 | | | | | | |
| | CGfamous | -0.05 | 5.71 | 0.43 | 0.13 | 3.46 | 0.46 | 0.06 | 3.3 | 0.03 | 3.5 | 5.7 | 0.01 |
| | CGccsm | 1.44 | 7.06 | 0.45 | -8.96 | 5.44 | 0.56 | 0.36 | 4.17 | 0.03 | -2.19 | 5.1 | 0.03 |
| Ant. | EBM | -1.73 | 4.01 | 0.75 | 16.04 | 10.46 | 0.56 | | | | | | |
| | CGfamous | -0.72 | 2.77 | 0.64 | -2.26 | 5.60 | 0.46 | 0.63 | 2.47 | 0.002 | 0.34 | 2.02 | -0.02 |
| | CGccsm | -7.3 | 4.57 | 0.53 | 3.19 | 8.14 | 0.55 | -1.16 | 2.41 | 0.06 | -0.09 | 1.87 | 0.15 |

**Table 9.** Mean Difference (MD), Correlation (COR) and RMSE relative to FAMOUS (except for last entry) over the ice region (Test period)

| | Model | TEMP (Deg C) | | | | | | PREC (cm/month) | | | | | |
| | | FEB | | | AUG | | | FEB | | | AUG | | |
| | | MD | RMSE | COR | MD | RMSE | COR | MD | RMSE | COR | MD | RMSE | COR |
| N. America | EBM | 1.68 | 4.24 | 0.99 | -0.83 | 2.21 | 0.93 | | | | | | |
| | CGfamous | -0.37 | 1.79 | 0.99 | -0.26 | 1.10 | .96 | 0.02 | 0.40 | .75 | -0.20 | 0.94 | 0.89 |
| | CGccsm | 1.02 | 3.2 | .98 | -0.68 | 1.94 | .94 | 0.02 | 0.31 | .77 | -0.08 | 0.78 | .88 |
| E.Asia | EBM | 0.14 | 1.65 | 0.99 | 0.03 | 0.20 | 0.65 | | | | | | |
| | CGfamous | 0.08 | 0.56 | 0.99 | 0.02 | 0.16 | 0.96 | 0.01 | 0.11 | 0.65 | 0.3 | 0.31 | 0.85 |
| | CGccsm | 0.5 | 0.40 | 0.99 | 0.14 | 0.90 | 0.97 | 0.002 | 0.07 | 0.43 | -0.01 | 0.18 | 0.78 |
| Anta. | EBM | -0.38 | 3.6 | 0.74 | 13.36 | 10.80 | 0.52 | | | | | | |
| | CGfamous | -0.4 | 2.57 | 0.66 | -1.48 | 3.96 | 0.45 | 0.35 | 0.5 | 0.008 | 0.40 | 1.28 | -0.04 |
| | CGccsm | -6.63 | 5.14 | 0.64 | 2.42 | 5.19 | 0.53 | -0.54 | 1.61 | 0.07 | 0.04 | 1.09 | 0.13 |





For North American precipitation over the test interval, the RMSE values of CGfamous are about 53% smaller (February: 1.9 versus 0.41 cm/month) and 31% smaller (August: 2.8 versus 4.1 cm/month) than the RMSE of CCSM precipitation (Table 6 and 8). Except for the weaker correlation for August compared to February (0.72 versus 0.92 cm/month) over the test interval (arguably again within structural uncertainty), CGfamous precipitation also passes this components of the climate Turing test.

Even with the increased complexity and higher resolution of CCSM, there is little deterioration in the training fit of CGccsm to CCSM compared to that of CGfamous to FAMOUS. All statistics (RMSE, MD and correlation) for CGfamous and CGccsm are better when computed just over ice covered regions (Table 7 and Table 9).

Our climate Turing test also requires assessment of map plot timeslices. A 100 ka August temperature fields comparison (Figure 7) indicates regional biases, with the most evident being a strong cold bias over Greenland. However, the discrepancies

are significantly less than the 18 ka difference between CCSM and FAMOUS (Figure 5 in the supplement). Furthermore, there is no obvious visual pattern that one could apriori use to ascertain which field was from FAMOUS versus CGfamous.

The geographic pattern of precipitation misfit between CGfamous and FAMOUS at 18 ka is very close to that of CCSM and FAMOUS (Figure 5 in the supplement). The misfit patterns for example instances from predictive and training regimes (100 ka and 18 ka in Figure 5 in the supplement) do not show obvious poorer predictive capability of the CG for the 100 ka timeslice

than for the 18 ka training timeslice.

For spatial comparison over time, the CCSM simulated fields and an ensemble of CGfamous simulations are projected onto the two leading EOFs of the FAMOUS climate model (Figure 9 for temperature and Figure 10 for precipitation over the North America). CCSM simulated fields have less variance compared to FAMOUS and the CGfamous simulated temperature field has a better fit with FAMOUS compared to CCSM. The time evolution of the expansion coefficients in the CGfamous simulated

fields compared to FAMOUS are comparatively better than that of CCSM. The 1st EOF represents about 56% of the simulated CGfamous temperature field, while the 2nd EOF represents about 15% (Figure 9). For the case of precipitation, only the first EOF is significant (37%), again with a closer match to FAMOUS than that of CCSM.

In summary, the CGfamous simulated temperature field better fits FAMOUS compared to CCSM versus FAMOUS and EBM versus FAMOUS and efficiently generates ensembles which represent large-scale climate variability. Even though the EBM has

no precipitation field, CGfamous is still able to capture FAMOUS precipitation fitted to our structural error (CCSM-FAMOUS) reference. Therefore the CG passes our Climate Turing Test.

## 6  Discussion

We have constructed a computationally efficient climate model (our Climate Generator or GCMs emulator) as an alternative to expensive GCMs for 100 kyr scale integrations. To compensate for inadequate variance in the BANN output within the CG,

Gaussian noise is injected into our CG at each time step with zero mean and spatially varying standard deviation calculated from ensemble networks prediction values (10% and 90% percentiles). The CG takes about 15 minutes to generate February and August climatologies over the 120 ka to 22 ka interval at 50 year time steps. To compare the prediction capability of our




(a) North Amerca (Full grid area)

(b) North America (Ice region)

(c) EurAsia (Full grid area)

(d) EurAsia (Ice region)

(e) Antarctica (Full grid area)

(f) Antarctica (Ice region)

**Figure 5.** Comparison of the spatial mean (with latitudinal weighting) February temperature time series on different continant. The black vertical line separates the test (left) and training part (right).



(a) North Amerca (Full grid area)

(b) North America (Ice region)

(c) EurAsia (Full grid area)

(d) EurAsia (Ice region)

(e) Antarctica (Full grid area)

(f) Antarctica (Ice region)

**Figure 6.** Comparison of the spatial mean (with latitudinal weighting) August temperature time series in different continant. The black vertical line separates the test (left) and training part (right).

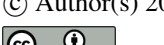



**Figure 7.** The August temperature field (Deg C) at 100 ka (1st and 2nd row) with elevation and ice contours shown in black and blue. The difference between plots are shown in the 3rd row. Model names are indicated in the top left corner in each box.

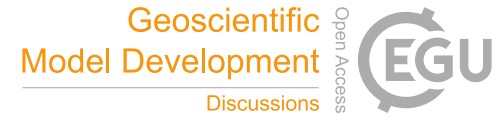

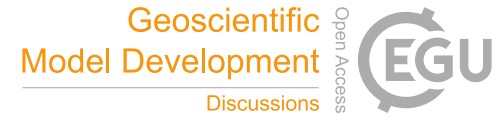

**Figure 8.** February precipitation field (cm/month). Left column: test case. Right column: Training Case. Model names and times are indicated in the top left corner in each box.





CG with the GCMs, we introduced "Turing test" concept as a "Climate Turing Test (CTT)". To implement the CTT concept, a direct comparison was done between CGfamous and FAMOUS Climate model outputs (120 ka to 22.05 ka). These data sets are not used in CG training. The difference between the CCSM and the FAMOUS over the training interval (22 ka to present year) are considered as a minimum structural uncertainty estimate for both GCMs. We take this uncertainty as a reference to

5   determine whether the CG passes the Climate Turing Test over the 120 ka to 22.05 ka test interval.

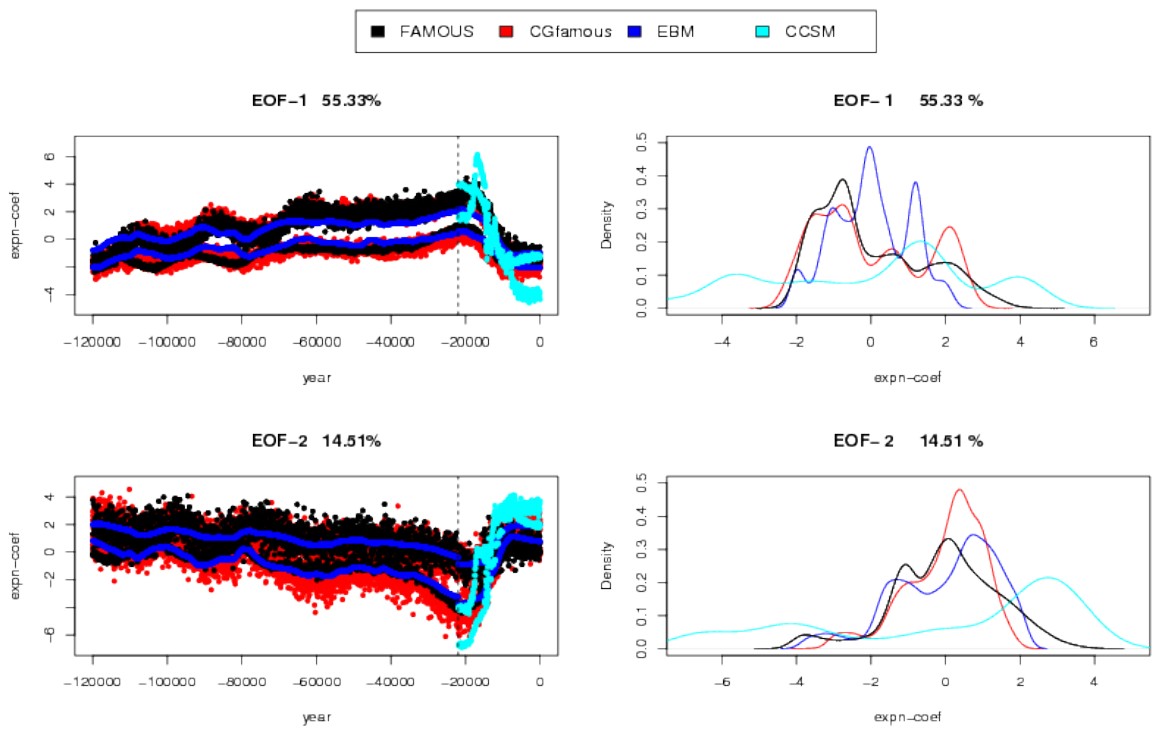

**Figure 9.** Temperature field over North America, left column: Distribution of the expansion coefficient over total time steps for the leading two EOFs of the FAMOUS Model data (black), and the distribution of the expansion coefficients over time obtained by an ensemble projection of CGfamous simulation (red), EBM temperature (blue) and CCSM (cyan) onto the same EOFs. Top time series represent the August EOFs and bottom time series are the February EOFs. Right column: Distribution of the expansion coefficients for the leading two FAMOUS EOFs for the FAMOUS (black), CGfamous (red), EBM temperature (blue), and CCSM (cyan) datasets

The CGfamous simulated fields have a smaller RMSE relative to FAMOUS (Temperature: about 50% less and precipitation about 33% less) in the test part (Table 4) compared to the CCSM uncertainty (RMSE) listed on (Table 3). CGfamous also has relatively better fits over the ice region. CGfamous extracts varying vertical temperature gradients given the significantly reduced misfits over high elevation regions compared to that of the EBM (Figure 7). The main comparative deficiency is the

10   significant dry bias over the Great Lakes (20 ka) and east thereof at 100 ka (Figure 8). The CGfamous simulated temperature





field has a cold bias (over the test interval) compared to FAMOUS (February and August) while CGccsm has a cold bias (August) and sharp warm bias (February) compared with FAMOUS over the test interval. A similar pattern of bias occurs over the training interval (Figure 5 and 6). The FAMOUS output has more variance compared to that of CCSM (Figure 5 and 6) in part due to the lack of available matched fields (only the radiative surface temperature from the CCSM and the 1.5 meter air

5    temperature from FAMOUS were available).

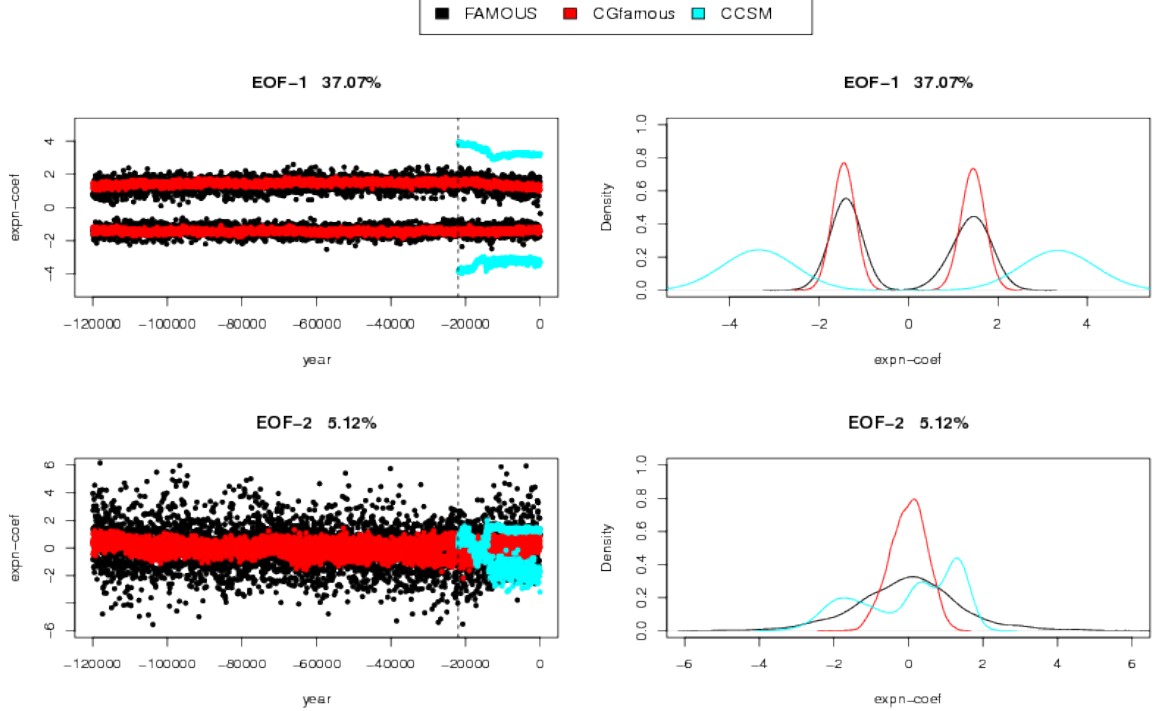

**Figure 10.** Precipitation field over North America, left column: Distribution of the expansion coefficient over total time steps for the leading two EOFs of the FAMOUS Model data (black), and the distribution of the expansion coefficients over time obtained by an ensemble projection of CGfamous simulation (red) and CCSM (cyan) onto the same EOFs. Top time series represent the August EOFs and bottom time series are the February EOFs. Right column: Distribution of the expansion coefficients for the leading two FAMOUS EOFs for the FAMOUS (black), CGfamous (red) and CCSM (cyan) datasets

The imperfection between climate generator predictions and reality can be conceptually broken down into two components. The first is the stochastic process error between the CG and GCM and the second is the structural error of the GCM relative to reality. Our simulated precipitation field has less variance compared to FAMOUS, and future development of the CG will explore other predictor sets which have relevance to precipitation prediction such as hydrology components.



# 7  Conclusions

We have introduced the concept of a Climate Generator to create a large spatio-temporal scale climate representation for coupled ice sheet modelling over glacial cycles. The CG expands the scale of weather generators. For this proof of concept, the CG was implemented over North America, northern Eurasia, and Antarctica. For validation, we compared CGfamous simulated fields against FAMOUS simulated fields (over the test interval which was not used for training the Bayesian artificial neural networks in the CG). We introduced the Climate Turing Test concept to provide a pass/fail reference for field comparison. The FAMOUS GCM was used for CG proof of concept/validation and then the CG was retrained against the much more advanced CCSM (CGccsm). CGfamous and CGcssm have test and training interval errors with respect to their corresponding GCMs that are of the same scale (and mostly less than) our minimal structural error estimate. This estimate is based on the difference between FAMOUS and CCSM temperature and precipitation fields. As such, the CG passes the Climate Turing test. It was not all a priori clear whether this would be possible given the CG reliance on the Energy Balance climate model. The CG will be coupled to the Glacial Systems Model (GSM) for experiments over the last glacial cycle. To simulate more atmospheric variables (like evaporation, etc.) the CG needs to be retrained with those GCM fields. In future work, the CG will be tested with full two-way coupling with glaciological models of all the major ice-sheets over the last glacial cycle. The CG approach will also be implemented and tested with more advanced EMICs (e.g. LOVECLIM) for shorter time scale contexts (given their increased computational expense).

*Code and data availability.*  CG testing output available upon reasonable request (concretely a maximum of 100 Mb). The BANN software is freely available : http://www.cs.toronto.edu/ radford/fbm.software.html. Scripts for generating CG BANNs available upon request.

*Author contributions.*  M. Arif wrote the initial draft of this manuscript and carried out much of the implementation and testing. T. Hauser started the project implementation. L. Tarasov conceived the "Climate Generator" and "Climate Turing Test" concepts, oversaw research design, and heavily edited this manuscript.

*Competing interests.*  The authors have no competing interests

*Acknowledgements.*  LT thanks Radford Neal for helpful discussion and for his original provision of the Flexible Bayesian Modelling toolkit. Both authors thank Robin Smith for provision of the FAMOUS model output.

This work was supported by a NSERC Discovery Grant (LT), the Canadian Foundation for Innovation (LT), the Atlantic Computational Excellence Network (ACEnet), and the Canada Research Chairs program (LT).



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
