# Peer review of "The Climate Generator: Stochastic climate representation for glacial cycle integration"

_Geoscientific Model Development, 2017_

## Short Comment (SC1) · 31 Jan 2018

Dear authors,

As explained in https://www.geoscientific-model-development.net/about/manuscript_types.ht

GMD is encouraging that authors upload the program code of models (including relevant data sets) as a supplement or make the code and data or the exact model version described in the paper accessible through a DOI (digital object identifier). In case your institution does not provide the possibility to make electronic data accessible through a DOI you may consider other providers (eg. zenodo.org of CERN) to create a DOI. Please note that in the code accessibility section you can still point the reader how to obtain the newest version.

If for some reason the code and/or data cannot be made available in this form (e.g. only via e-mail contact) the "Code Availability" section need to clearly state the reasons for why access is restricted (e.g. licensing reasons).

Additionally, please note, that our Editorial version 1.1 (http://www.geosci-model-dev.net/8/3487/2015/gmd-8-3487-2015.html) highlights the requirements of papers published in GMD. In particular your paper does not meet the requirement that the version number or other unique identifier of the model described in the manuscript must be included in the title. Therefore please add the version number of the climate generator described in this article to the title upon revision of your article.

Best regards, Astrid Kerkweg

---

## Author Comment (AC1) · 31 Jan 2018

As previously indicated in our reply to the editor, this is a proof of concept paper and not a model description paper. Therefore it is meaningless to provide a version number.

For proof of concept, archiving 10's of Gb, doesn't make sense and therefore we believe are code availability statement is appropriate: < \codedataavailability{CG testing output available upon reasonable request (concretely a maximum of 100 Mb). The BANN software is freely available : http://www.cs.toronto.edu/~radford/fbm.software.html. Scripts for generating CG BANNs available upon request.}

As we progress, I suspect the scripts will be revised and slowly made more user friendly, so making them available up request makes more sense than archiving the

current not so user friendly research scripts.

---

## Referee Comment (RC1) · M. Crucifix (Referee) · 16 Mar 2018

**1   Summary**

The purpose of this article is to provide the rationale and documentation for a stochastic *climate generator* developed by the authors. The climate generator is stastical model based on a Bayesian neural network architecture, which provides gridpoint-wise climate outputs from a tuple of predictors (their Figure 1) which includes elevation, ice cover, greenhouse and orbital forcings, as well as the temperature predicted by an energy balance model. The model is trained on a series of experiments with two general circulation models: FAMOUS and CCSM. The purpose of the *climate generator* is to be coupled with an ice sheet model to simulate glacial-interglacial cycles. The

assessment of this *climate generator* is based on a series of metrics and benchmarks, which the authors present as a "Turing Test". The assessment relies mainly on the comparison of the model outputs (trained on FAMOUS) to simulations with FAMOUS for configurations in the training set.

**2  Overall comments**

2.1   Justification of the model

The 'Climate Generator' may actually be seen as a sophisticated machine-learning-based correction of energy-balance-model (EBM) outputs. Compared to other meta-modelling strategies proposed so far uses (as input) the the ice boundary conditions provided grid-point-wise. It does not need to rely an ice volume index aggregating the state of ice sheets at the global level.

The purpose of the introduction is to justify this effort, and it may fall a little bit short of this.
On the one hand, there is a dissonance between the claims of the introduction and what the paper actually offers. For example, the introduction explains that "Regression-based methods are relatively straightforward [. . . ] but have an inadequate represen-tation of observed variance and extreme events". Does the climate generator address these shortcomings (see below, remarks on variability)? The authors also explain that weather generators "replicate the statistical attributes of local climate variables [re-viewer note: wouldn't it be *weather*? ] rather than the observed sequence of events". In what sense would the "climate generator" *replicate* the statistical attributes of "cli-mate" variables? Later in this review, I raise some questions about the lack of consis-tent representation for decadal or centennial modes of variability. Isn't it indeed what one would have expected from a "climate generator" ? Finally, more needs to be said

about how the climate generator will actually contribute to the grand challange of deciphering the mechanisms of glacial-interglacial cycles (if this is indeed what the authors are after).

**2.2 The methodological implementation**

**2.2.1 BNN architecture**

Crafting a BNN architecture is an art which, clearly, the authors master much better than this reviewer. The paper is thus an opportunity to introduce the reader to that art. We need to understand better the elements that makes the BNN designer opt for a certain architecture, when the objective is to emulate a climate model. As its stands the article features different architectures and compares their performance, but the main message is somehow diluted into the numerous tables and graphics, some of which are a little bit cryptic. In the end, I was left with what seems to be the key questions: why the authors originally opted such or such architecture, and how one could interpret the fact that some architectures seem to be working better than others? The authors should read this criticism as request for less rather than more content in the main article (authors are still free to put long tables in S.I.). At the end of the article, the reader should be satisfied that she understands the critical aspects in the development of BNN models for climate simulators, in order to be able to replicate the effort possibly with other climate simulators.

**2.2.2 Modelling variability**

The hypothesis (which the authors present as an assumption or approximation) that the predictive uncertainty of the BNN is in good part "due to the internal variability of the GCM climate" is certainly the most controversial technical aspect of the article.

There is in fact little to give substance to that hypothesis. As I understand it, the BNN (associated with the EBM) is only but an approximation of the GCM, and it is very unclear why the misfit would actually match the GCM internal variability. To mitigate this criticism, it must be observed that a regression model calibrated on GCM experiments may sometimes predict the model stationary mean so well, that the misfit between the predicted mean and a specific GCM experiment is mainly due to the finite sample length of the GCM experiment. Some have seen this before (Araya-Melo et al., 10.5194/cp-11-45-2015). However, turning this 'misfit' into a model for interannual or interdecadal variability requires some careful thoughts and discussion. The fact that, in the validation procedure, the difference between FAMOUS and CCSM is further used as an upper bound for the climate generator misfit (and thus, if I understood correctly, the acceptable level for self-inferred uncertainty) makes it even more confusing.

On the other hand, the modelled climate variability is assumed to be spatially uncorrelated, a point over which the authors do not seem to worry much about because ice sheets would integrate perturbations over long times. This defence is disputable. Spatial variability patterns such as those active in the ocean at the decadal and centennial scales may have specific and interesting consequences for the development of ice sheets. As suggested in the introduction of the present review, one could have argued that a *climate* generator should actually be a model which, compared to a weather generator, features carefully the structure of spatio-temporal decadal and centennial variability modes. For the same reason, the choice of a Gaussian noise is *a priori* arguable (as it may not represent the important consequences of extreme events).

**2.2.3 Training**

The authors describe the training procedure but say little about how the training can be made efficient. How many experiments are needed, how the choice of training simulations could be optimised, and to what extent the test cases used for the validation of the climate simulator are convincingly / sufficiently independent of the training experiments.

**2.3 Turing test**

With regret I must confess that I found the use of the "Turing test" phrase more obfuscating than enlightening. The Turing test was imagined in the context of artificial intelligence, where self-reference and induction are important concepts. In the present context the evaluation is essentially a benchmarking process, the output of which could best be summarised with a colorful table witch checkers, crosses, indicating clearly which criteria are met and which are unmet.

**3 Other comments**

- The tables and graphics are not always self-explanatory (especially the Taylor diagrams). Honestly, I found that this was a hard paper to read. Some editorial work is needed to streamline the paper.

- Section 2, which is currently oddly structured (one page of material at the section level, then section 2.1 subdivided into four subsections) could be more focused, with less emphasis on meta-digressions such as "reasoning behind the name of CG". I suggest that both more impact and a better reading experience could be achieved by focusing on what is needed to simulate glacial-interglacial cycles, and how the stochastic climate generator will contribute to our understanding of glacial-interglacial cycles.

- The material under section 4.2 could best be illustrated with a flowchart. Make sure that all concepts are sufficiently explained to the non-experts (e.g. "hybrid

Monte Carlo sampling"). Among others, clarify concepts such as "sub-group of parameters" and "shape parameters".

- p.8 : How do you "get an idea" of when the simulation has reached equilibrium. Can't you use a formal criteria?

- p. 14 is quite descriptive. More effort could be given to extract the key message. The interest of the numerous tables should also be reconsidered.

- Insolation means "Incoming solar radiation". Avoid "solar insolation". In passing, why are the authors using 60°N insolation, rather than insolation predictors ($e \sin \varpi$, obliquity) which are not bound to a specific latitude?
* * *

---

## Referee Comment (RC2) · L J Gregoire (Referee) · 19 Mar 2018

L J Gregoire (Referee)

l.j.gregoire@leeds.ac.uk

This manuscript present the development of a novel tool to efficiently model climate change during long time period such as glacial interglacial cycle. This is a statistical tool which "generates" climate based on temperature simulated by an efficient climate model (here an Energy Balance model; EBM) and other parameters (geographical coordinates, elevation, greenhouse gases, orbit...). This "climate generator" is trained on long simulations run with two general circulation models, FAMOUS and CCSM3. This work is very novel and interesting and this manuscript is a perfect fit for the GMD journal. I would recommend this manuscript for publication within GMD, but with some more discussion and improvement of how the methods and results are presented. I think broadly the work is robust and sound and the analysis is complete.

Main comments:

One important aspect of this tool is that it requires input of temperature from an EBM as well as being trained on output from a GCM. I think this needs to be further explained. In a sense, this tool provides bias-correction to the EBM as well as incorporating climate variability (in time and space) that is learned from the GCM and providing compatible precipitation. I would be interested to know why EBM temperature input is required, would it be possible to train a tool that just requires coordinate, time (or GHG, orbit) and elevation input. More broadly, I think clarifying the assumptions and aims of this climate generator might help describe the methodological choices made and the criteria for validation.

The introduction section on page 3 reads like a long list of methods that could have been used for this study. I found it a little bit hard to follow (lots of new concepts for me). Can you clarify why specific method aren't used here, highlight a bit more which one is used and why ? I think all the information is there, but a few tweaks would help the reader assimilate the ideas.

The text needs a bit of reworking. I would suggest section 2.1 be merged with section 3 and 4 as it all describes "Methods". Section 2.1 by attempting to be just a part of the introduction ends up being too vague in places, particularly for 2.1.4 Climate turning test. Section 4.2 on the BANN implementation is a bit hard to follow for a non-novice, I think this section requires a bit more narrative and explanation of the process.

The results section is also hard to follow because the tables and figures are not systematically referenced in the text. I'm not even sure that all the figures are referenced in the text.

The discussion section seems to end a bit abruptly with the introduction of the concept of "reality". In this study, "reality" is whatever the climate generator is trained on. This is kind of explained in section 2.1.3, but not so clearly. I would suggest clarifying to justify the comparison made between CGfamous and FAMOUS and expanding the

discussion section on this topic.

On a related note, there is a lot of comparison of CGccsm with FAMOUS, but is that comparison fair ? If the climate generator is trained on data from CCSM, its effectiveness should be tested against CCSM (the new "reality"). I note in particular that CGccsm is consistently better at matching CCSM than CGfamous is a matching FAMOUS. Could you comment on this point and suggest a reason for this ?

5 consecutive years of FAMOUS isn't quite a 50 year climatology despite the acceleration. The interannual climate variability is not accelerated in the model, therefore an average of 5 years will not give a climatology equivalent to a mean of 50 years. This has important implications for the definition of climate "noise"/variability in this study and for the behaviour of the simulator. Please include a discussion of this in the manuscript, with specific reference to what timescale of climate variability the stochastic noise added is meant to represent.

The testing/validation of the climate generator is done only for two months of the year, February and August. Why not do the RMSE and Taylor diagrams on all months of the year aggregated into one metric ?

More detailed comments:

Page 5 line 5 and line 19. I think those two sentences say the same thing.

Page 5 line 6. Please develop a bit more why these EOF in theory could help capture non-local effects and what you mean by that. I think I can guess, but it is not that obvious.

Section 2.1.4. quantify what is meant by "relatively high correlation,[. . .] close patterns and reasonable capacity. . ."

Section 3. I think there is some confusion between training period and test period. CCSM3 is only available over the training interval (deglaciation).

Page 6 Line 19. Justify choice of lapse rate and implication

Page 6 line 29 "unreasolved variability" by what, the EBM ?

Page 9 line 26 the last sentence in this paragraph is a bit confusing, what you are doing is comparing the results of BANN with a different set of inputs and architectures.

Table 1. What is the difference between "CO2+ CH4" and "carbon dioxide, methane" in PS2 and PS3 ? are you using the sum of greenhouse gases in PS2 are are they two individual inputs ?

Table 3 and onwards, can you clarify that the MD and RMSE is aggregated not only over space, but also over time.

Page 14 line 2, "Table (4)" -> Table 4

Page 15 line 8, please justify why you take the difference between CCSM and FAMOUs as the minimum value of model uncertainty.

Page 15 line 11 (and elsewhere throughout the results), refer to appropriate table.

Page 16 line 4 "test part" -> test period ?

Page 16 line 11 TABLE -> Table

Figure 5, you have two blue line that are hard to distinguish

Figure 6. Can you comment on the fact that over the ice region, the prediction of spatial mean august temperature over the north American ice sheet by CGfamous are radically different (and appear to the anticorrelated) to both FAMOUS and the EBM tempreatures.

Figure 7,8 the use of column is confusing in a landscape figure, I would just make more use of labels and state dates in the captions

Page 25 line 8 CGcssm -> CGccsm

Page 25 line 13. What do you mean by "retained" ?

Page 25 Line 15: "more advanced" than what ?

Repetition within some of the figures in the supplementary material. For example between figure 6 and 7. The supplementary figures do not all follow the same template, which makes them confusing to read.